# 'Wish you were here': Managers' experiences of hybrid work in higher education

Helena Tinnerholm Ljungberg[1]*, Martina Wallberg[1¤], Emmanuel Aboagye[1,2], Gunnar Bergström[1,3], Christina Björklund[1], Lydia Kwak[1], Susanna Toivanen[4], Irene Jensen[1]

1 Unit of Intervention and Implementation Research for Worker Health, Institute for Environmental Medicine, Karolinska Institutet, Stockholm, Sweden, 2 Department of Psychology, Norwegian University of Science and Technology, Trondheim, Norway, 3 Department of Occupational Health, Psychology and Sports Sciences, Faculty of Health and Occupational Studies, University of Gävle, Gävle, Sweden, 4 School of Health, Care and Social Welfare, Mälardalen University, Västerås, Sweden

¤ Current address: Department of Economic History, Uppsala University, Sweden
* helena.tinnerholm.ljungberg@ki.se

## Abstract

The prevalence of telework increased dramatically during the COVID-19 pandemic, and today it is not uncommon to refer to hybrid work as "the new normal" in work life. Leadership plays a pivotal role in hybrid work transitions, underscoring the need for research on post-pandemic managerial practices. This qualitative interview study with 15 professional service managers at a Swedish medical university, working in either central administration or a research department, provides a nuanced understanding of the experiences of implementing hybrid work in a higher education setting. The qualitative content analysis resulted in three main themes and six sub-themes: New ways of organising work (sub-themes: Hybrid work brings new opportunities and needs, and Hybrid work as an ongoing process of change); Changes for employees (sub-themes: Social interaction and sense of community, and Increased work-life balance); and Changes in leadership (sub-themes: Communication with employees and New expectations on managers). The findings of this study provide a more fine-grained understanding of how managers experienced both challenges and opportunities in implementing and managing hybrid working arrangements. Challenges included managing employee expectations and relations, while opportunities included potential improvements in work-life balance. A key conclusion of this study is that managers in hybrid work environments adjust their leadership, especially when communicating and managing relationships within teams and across the organization. Despite the identified challenges and despite managers' wish to see their employees in person and on site, the interviewed managers are generally optimistic about hybrid work and see it as the future. To address the identified challenges, managers may benefit from networking and exchanging information with other managers in similar situations, as well as support from their organisation.

**Data availability statement:** The datasets generated and analysed during the current study are not publicly available due to the Swedish ethical review regulation. Inquiries for data access should be sent to Karolinska Institutet, Institute of Environmental Medicine, Unit of Intervention and Implementation Research for Worker Health, Box 210, 171 77 Stockholm, or contact the principal investigator Irene.jensen@ki.se, who will then contact the Swedish Ethical Review Authority, https://etikprovningsmyndigheten.se.

**Funding:** The author(s) received no specific funding for this work.

**Competing interests:** The authors have declared that no competing interests exist.

## Introduction

The prevalence of telework, i.e., work partially or fully performed outside the formal workplace, increased dramatically during the COVID-19 pandemic [1,2]. The slow and steady trend towards an increase in telework had already started before the pandemic, not least in Scandinavia [1,3–5]. This development was facilitated by the development and increased use of digital infrastructures, in which high-speed broadband and digital tools for video conferencing and file sharing played a vital role [2]. However, it is important to distinguish between pre-pandemic telework, partially responsive to individual preferences [6] and organizational norms, and telework during the pandemic, which was imposed by national restrictions regarding mandatory work-from-home [7] and affected by the emergency context of the pandemic [8]. It is particularly important to acknowledge this difference, as there are unique challenges pertaining to leadership and managerial practices in post-pandemic telework [9].

The term *hybrid work*, which is used in this article, is commonly used in research following the COVID-19 pandemic to describe a situation where work is conducted both on-site and online from home or other physical locations [8]. Other terms, such as partial telework, remote work, and work from home, are also used interchangeably with hybrid work in some contexts, even though it has been argued that hybrid work differs conceptually from other forms of telework arrangements [10].

Before the pandemic, virtual teams, i.e., groups of employees working together across different locations, were not seldom global, which in some cases created challenges for managers in supervising staff in different time zones via asynchronous digital communication [6]. During the pandemic, it was not uncommon that teams that were used to seeing each other on site every day turned into virtual teams, as was the case in the setting studied in this article. However, after the pandemic, hybrid work has become increasingly common. As Barrero et al. show, almost a third of all full-time workers in the United States had a hybrid work arrangement in 2023 [2].

There are lessons to be learned from previous studies of telework concerning the differences between telework before, during, and after the COVID-19 pandemic. Previous research has shown that telework can lead to several positive effects in the workplace, such as reduced commuting time [1] and maintaining employee efficiency [11]. Other positive effects include a potential increased sense of control over the work situation for employees [11,12], higher self-reported productivity for both employers and employees, [13] alongside less intention to leave and employee turnover [13]. Some research points to benefits related to increased work-life balance [7,11], while others highlight that teleworking might have negative effects on work-life balance and well-being, not least for women [14].

Despite the positive effects of telework found by previous studies, neither full-time telework during the pandemic nor hybrid work after the pandemic is without challenges. The negative effects of telework identified during the pandemic, such as blurred work-life boundaries [15,16], social isolation [12], and reduced knowledge sharing across the organization due to more static communication

networks [17] might remain even after a transition to partial telework or hybrid work after the pandemic [6]. However, post-pandemic hybrid work arrangements have been highlighted as a way of addressing feelings of social isolation in full telework [4] while also facilitating employee retention and performance [18]. The proposed benefits of hybrid work arrangements have led some to describe hybrid work as the future of working life [7]. Although hybrid work can positively influence working life, this may not always be the case. Studies from different contexts have highlighted the importance of acknowledging individual differences when examining the effects of hybrid work [7,19]. These studies have shown that the transition to hybrid work arrangements requires the consideration of contextual and organizational conditions on multiple levels [19–22].

The post-pandemic literature examining the transition to hybrid work has predominantly focused on the viewpoints of employees with few studies exploring the perceptions of managers during this transition, despite them being described as key stakeholders in making the transition to hybrid work successful [7,19,20]. An interview study with managers from public and private organizations in Sweden found that managers viewed teleworking as beneficial for their organization as it strengthened recruitment and retention efforts and positively influenced team dynamics [23]. However, telework, including hybrid work arrangements, has been found to bring challenges for managers. A study by Kirchner, Ipsen and Hansen [24] showed that managers reported more challenging work conditions than employees due to redefined work tasks and social isolation. Other studies have highlighted challenges with virtual management practices as they require new technical skills and adaptations from managers [20,25] and creates new challenges for managers to sustainably lead and support their employees [25,26]. Studies have shown that virtual managerial practices differ from traditional managerial practices as they involve a change in trust and communication in the relationship between managers and employees, as well as between managers and other organizational stakeholders [20]. One can argue that in a hybrid work setting, the managers have to manage both virtual and traditional managerial practices. This change in leadership conditions and performance can imply an increased need for managerial support from the organization [27].

Given the importance of managers in the transition to hybrid work arrangements, more research on post-pandemic managerial practices and leadership is needed [6,28]. Moreover, research in this field benefits from increased concern given to contextual and organizational factors, and managers' perceptions of hybrid work in various sectors, not least in higher education institutions (HEIs). In Sweden, when taken together, HEIs were in 2022 the largest employer in the governmental sector, entailing 26 percent of all governmentally employed [29]. Statistics from the Swedish Higher Education Authority show that among the 81,000 employed by a HEI in 2024, approximately 18,000 fall under the category of administrative personnel, 7,000 as technical staff, and 1,000 as librarians [30]. These categories, however, do not completely capture the shift that has happened in Sweden, as well as internationally, with regard to the increase in both education level and competencies among professional support staff in the "third space" at HEIs [31,32]. This third space can best be described as "*an emergent territory between academic and professional domains,*" which has emerged to meet new demands [33 p. 377.]. A study on staff composition in Danish universities over time concluded that the increase of highly educated professional staff seems to coincide with an increase in "relatively low-wage temporary positions" among academic staff [34 p. 629.], and that this trend is also seen in other national contexts. Highly educated professional support staff in Sweden also report that they have a strong influence over the design of their work roles and the work they perform [32]. The multitude of positions and work tasks performed under the umbrella of professional services at HEIs adds to the challenge of being a manager in this setting.

In the context of HEIs, previous research on hybrid work has mainly focused on teaching and researching staff [35]. This study aims to address the above-mentioned knowledge gap by providing a nuanced understanding of professional service managers' experiences of implementing hybrid work in a higher education setting.

## Methods

The consolidated criteria for reporting qualitative research (COREQ) are followed in this study [36].

## Setting and procedure

This study is part of a comprehensive evaluation of hybrid work among employees and managers working within the professional services at a Swedish medical university. This medical university has professional services employed both at various research departments and in a centralized administration. Moreover, the employees work from multiple locations (on two main campuses and several hospitals). In September 2021, hybrid work was implemented in several government agencies, including higher education institutions in Sweden. The current study focuses on managers' experiences of the transition from having employees working almost exclusively on-site before the pandemic, working extensively from home during the pandemic, to a hybrid work arrangement after the pandemic in which employees combine on-site work with remote work. Hybrid work at the university was defined via a policy that allowed employees to work from another location up to 49 per cent of their working hours if it was deemed compatible with the needs of the department, which was discussed between the employees and their first-line manager.

The transition to hybrid work was prepared for at the university during fall 2020, when national restrictions in Sweden recommended employees to work from home as much as possible. These national recommendations were lifted in September 2021 and the implementation of hybrid work arrangements started at the university. During a short period (December 2021 to February 2022) the national recommendations were reinstated, and the implementation was put on pause. In March 2021, the university decided to fund an evaluation project to examine both managers' and employees' experiences of the work situation and health in relation to the transition to hybrid work.

## Design

An interpretive qualitative study design was used to explore managers' perceptions of the transition to hybrid work, as it is suitable for exploring nuances in experiences and perspectives among stakeholders [37]. A semi-structured interview guide was developed based on the aim of the study and a review of the literature on the effects of the COVID-19 pandemic on working life. The interview guide covered the following themes: previous and current experiences and perceptions of hybrid work; implemented changes in the work group; experiences (including challenges and benefits) of being a leader in a hybrid work setting. The guide was pilot-tested in one interview with a professional service manager at the university by two of the authors, HTL and MW, who then discussed the form and outcome of the interview. No changes to the guide were deemed necessary before proceeding with the interviews.

## Data collection and participants

Research participants were recruited and interviewed between May 23 and July 27, 2022. A purposive sampling strategy was used to get a varied sample in terms of gender, campus location, place of work (central administration; department), and focus of work (research- or teaching-intensive departments). Based on a list of administrative first-line managers and middle managers at the central administration and the research departments (N=66), invitations were sent to 17 managers, of whom 13 agreed to be interviewed. Additional invitations were sent to five managers, of whom two agreed to be interviewed. Throughout the data collection process, the interviewers engaged in an ongoing discussion about the collected data. They assessed how each interview added new perspectives or enhanced their understanding of previously mentioned experiences. This discussion guided the number of interviewees, and data collection was completed after 15 interviews, which were deemed sufficient to meet the study's objectives. Table 1 provides a descriptive overview of the study sample.

Authors HTL and MW, both experienced in semi-structured interviews, conducted eight and seven interviews respectively. The interviewers had no prior relationship with any of the interviewees. The interviews lasted approximately 50 minutes (min. 35 and max. 76 minutes) and were conducted either on a digital platform (Zoom or Teams) (n=13) or in person (n=2), based on the participants' preferences. The purpose of having the interviewees choose the location was to make

**Table 1. Participants descriptives.**

|  | Managers |
| --- | --- |
| **Number** | 15 |
| **Gender** | |
| Male | 7 |
| Female | 8 |
| **Years of experience in the current manager position** | |
| ≤ 4 | 9 |
| ≥ 5 | 6 |
| **Number of subordinates in the managers' work group** | |
| ≤ 20 | 6 |
| ≥ 21 | 9 |
| **Organizational affiliation** | |
| Central administration | 8 |
| Department | 7 |

the interviewees feel comfortable [38]. In the digital interviews, both parties used the video function to make it possible to observe the body language [38]. The interviews were recorded with the interviewees' consent to facilitate the forthcoming data processing.

## Data analysis

All 15 recorded audio files from the interviews were transcribed verbatim. Comparisons between the audio files and transcripts were made to ensure quality. The material was then analyzed using a qualitative content analysis, and more specifically, the conventional content analysis approach as described by Hsieh and Shannon [39]. In the conventional approach applied in this study, categories are developed during data analysis and derived from the analyzed text, providing researchers with a deep understanding of the phenomenon under study. This analysis method is therefore particularly suitable when the studied phenomenon is relatively unexplored [39], which is the case with managers' experience of hybrid work after the pandemic.

Following the conventional content analysis approach [39], the analysis was conducted as follows. All interview transcripts were read in full to gain a comprehensive understanding of the material. MW initially coded four interviews in Microsoft Word to maintain an overview of the material and preserve proximity to the empirical data. For validation purposes, the initial coding was discussed between MW and HTL before the remaining coding was done. The four coded transcripts were read thoroughly by HTL, and notes were made when interpretations differed. The notes were then discussed until a consensus was reached. Some changes and additions were made, and the coding of the four interviews was then entered into the software NVivo 11, where the remaining eleven interview transcripts were coded and sorted into categories and subcategories by MW. All codes were examined in the analysis, and similar codes were grouped to form subcategories and main categories. To strengthen the credibility of the study, the coding and interpretations of the results were discussed first by MW and HTL, then by all authors (HTL, MW, EA, IJ, GB, CB, ST, LK) until a consensus was reached.

## Ethics

This study was approved by the Swedish Ethical Review Authority (registration number 2021–03637). All participants received written and oral information about the research project before giving informed consent (either written or verbal). The participants who gave their consent verbally did so at the beginning of the interviews, and this part was also recorded.

## Results

The analysis resulted in three main themes and six subcategories, as presented in Table 2. The three main categories were: new ways of organizing work, changes for employees, and changes in leadership.

### New ways of organizing work

The following category reflects the managers' perceptions of how work in their teams has been affected by implementing hybrid work: what opportunities and needs it entails, and how the transition towards hybrid work is to be understood as an ongoing change process.

**Hybrid work brings new opportunities and needs.** Managers often compared pre-pandemic work processes (mainly on-site with some degree of work from home) with post-pandemic hybrid work, stating that the new ways of working included having better digital tools and new digital competencies achieved during the pandemic. The managers reported different levels of experience of being managers in a hybrid work setting prior to the pandemic (from work in other organizations), which influenced their attitudes towards hybrid work. Managers who were more experienced or familiar with hybrid work expressed, in general, a more positive attitude towards the idea of implementing hybrid work than those with less experience or familiarity with hybrid work. However, the attitude towards hybrid work among the interviewed managers who reported that they had initially been skeptical of it during the pandemic changed over time as they experienced that their employees managed to stay equally productive.

I like this flexibility. Compared to how rigid I was from the start, I have really changed. Because I see the advantages of this [the hybrid working arrangement], and I think it works so well. And I like this freedom under responsibility. And when you have employees like I do, who are incredibly loyal and ambitious, it's no problem. (IP 12)

The managers described how recommended telework during the pandemic placed a demand on both them and their employees to quickly adapt to the increased use of digital tools. A perceived benefit of this digital adaptation was that it paved the way for quicker and more effective ways of supporting employees:

Now it has become much more common with 'share screen' and 'show in the system', and it has become much easier. So, it has really... We provide better support today than before the pandemic. (IP 3)

**Table 2. Categories, subcategories, and descriptions of subcategories.**

| Main categories | Subcategories | Description of subcategories |
|---|---|---|
| New ways of organizing work | Hybrid work brings new opportunities and needs | Managers' views on hybrid work have changed after the pandemic, with many now recognizing new opportunities associated with a more digitized way of working. Reflections on which new needs managers identify concerning hybrid meetings, office spaces/design, and technological equipment. |
|  | Hybrid work as an ongoing process of change | New ways of working have not reached a static form but are evolving, with examples of different solutions to establish novel ways of working. At times, a process that included feelings of uncertainty connected to expectations of on-site presence. |
| Changes for employees | Social interaction and sense of community | Social interaction with and between employees has changed. New challenges related to being a new employee or maintaining employee relations. |
|  | Increased work-life balance | Hybrid work can lead to increased balance and make it easier for employees to combine work and personal life. |
| Changes in leadership | Communication with employees | The new way of working has affected communication between managers and employees and entails benefits and challenges. |
|  | New expectations for managers | New expectations from others in the organization, employees, and the managers themselves have affected the manager's role. In addition, hybrid work has led to new and changing demands on the manager's work. |

The redefined work processes and increased flexibility for employees to organize their work in a hybrid work arrangement were also perceived to have strengthened the organization's competitiveness in the labour market and to facilitate the recruitment of new employees.

When you are looking for a job now or thinking about changing jobs, flexibility is almost as important as the salary. [...]. So, I think, as an attractive employer, we have to offer flexibility; otherwise, we will have difficulty recruiting. (IP 13)

Managers also identified new challenges due to hybrid work, relating to the use and viability of digital tools. Holding hybrid meetings, in which some participants participate on-site while others participate digitally, was highlighted as particularly challenging. The difficulties in holding hybrid meetings without complications, which placed additional demands on the person leading the hybrid meeting, led several managers to opt for either digital or on-site meetings:

If we have a good technology platform where it is easy to have hybrid meetings in that format, then it would be okay. But as things stand now, I am more in favor of one or the other. (IP 1)

The managers described how hybrid meetings required technical conditions that were not always available or user-friendly. Furthermore, some managers reported that hybrid work had triggered discussions about the need for new office designs that would provide opportunities for the increased use of digital meetings at the office. From the manager's perspective, some meetings required a physical presence to achieve their purpose:

It is more this, the time that we have that is not spent on just our basic mission. The one that becomes... yes, but development, coming up with new ideas, innovation or... what becomes in a discussion, [...] in a spontaneous meeting where you hear something and "yes, but that is what I am doing here" or you can get some synergy effects. We are losing that now. (IP 3)

This was particularly the case with meetings for discussing new ideas and initiating creative processes.

**Hybrid work as an ongoing process of change.** The managers indicated that the hybrid work arrangements were still under development. Several managers had ongoing thoughts about how to, in practice, organize the distribution of telework versus work in the office for their employees. In addition, there were varying degrees of control over when employees were expected to be on site. Several managers had a fixed office day per week for themselves and their staff, while others described how decisions on the distribution of attendance were made through regular evaluations with employees. However, the managers agreed that it was ultimately the needs of the organization that determined attendance:

The planning on... with the return, we said "then we will have fixed days", we thought that was great, and so on. But then... only a week went by and we felt [...], why should we have that? Sitting in meetings when I am here, then I will not meet anyone anyway [...]. Then we said, we should set the schedules as we want, the only thing we have to think about is that there should be [someone] on site every day. (IP 11)

A couple of managers experienced some uncertainty among employees about how time spent working from the office and telework should be distributed, especially if their employees felt that the expectations of attendance were different for different employees or employees in other parts of the organization. On the one hand, the managers considered it essential to be transparent in decisions about the requirements for attendance, and on the other hand, they wanted to protect the integrity of employees in cases where an employee, for personal reasons, teleworked more during a period. Some of the managers emphasized transparency towards employees in decisions relating to hybrid work.

Even when managers highlighted challenges with the hybrid work arrangement, they often concluded that it was the working arrangement they saw also for the future.

> ... Just like this, hybrid work is the way of the future, but we mustn't hide the fact that we also have to work on the challenges of the hybrid workplace, because they do exist. (IP 13).

Even though the hybrid work arrangements were regarded as an ongoing process and under development, it was seen as here to stay.

### Changes for employees

This category describes how the managers perceived the impact of hybrid work on employees regarding changes in social interaction with and between employees and employees' work-life balance.

**Social interaction and sense of community.** Several managers described how the social interaction in hybrid work had changed since they no longer met their employees and colleagues daily. Managers also reflected on maintaining a sense of belonging to the group when the spontaneous interactions in the coffee rooms and corridors were no longer a natural part of the working day. There was also an expressed concern that the decrease in social interaction could, in the long term, affect employee engagement by weakening the sense of community within the team.

> I mean, the "we" feeling in the unit, that we are almost never all on-site at the same time, but there is always someone who is... And if you are on-site, […]. Some kind of feeling is built up, and I think it takes longer to build it up if some people are always working from home several days a week. (IP 8).

As part of bringing employees back to the office after the COVID-19 pandemic, there was an increased emphasis on activities encouraging social interactions. Such activities were described as necessary to build and maintain a common culture in the team, and some managers argued that a new and vital part of the managerial role had become encouraging employees to make space for social interactions at work.

Another challenge related to social interaction in hybrid work was identified as particularly prominent for new employees:

> The social aspect is very difficult for someone who is new. […] many of my employees have worked together for a long time before the pandemic, so it's easy [for them] to maintain social interaction online, via chat, and to keep in touch by phone and, to some extent, see each other in the digital space. (IP 1)

Furthermore, getting into the group and getting to know one's colleagues as a new employee or even a new manager was considered a more time-consuming challenge in hybrid work.

**Increased work-life balance.** The increased flexibility was considered to have led to less employee stress, as illustrated by one manager:

> These new, more flexible times that you get because you don't have to go to a workplace physically, means that you have more time in the morning or it's not as stressful. [...], that you can get the everyday puzzle together much better (IP 4)

Most managers perceived that overall employee well-being had increased due to better work-life balance and decreased time spent commuting.

> And most people feel a responsibility to do their job and deliver results. And it doesn't matter, they do it if they are sitting at home too, or if they are sitting somewhere else. If they can receive a fridge that's delivered, or if they can go out for a

run at lunch or exercise, that's much better. Because I believe that employees feel better and deliver better if they make sure they are well. (IP 6)

The ability to choose to work from home to balance everyday life was highlighted as an advantage of the new way of working, which in turn had increased employee satisfaction. Even on the days that the employees worked in the office, the flexibility contributed to a better balance by providing the opportunity to go home earlier to pick up children from preschool, for example, and then be able to catch up on work in the evening.

## Changes in leadership

This category presents the managers' experiences of how their leadership role and tasks has been affected by the transition to hybrid work and subsequent changes in communication with employees and new expectations and demands in the leadership role.

**Communication with employees.** When reflecting on how their leadership had been affected by the hybrid working arrangement, some managers said that it had become easier to get in touch with their employees and colleagues quickly:

And then it is... amazingly much easier now than it was before, when you had to synchronize. "Are you there then? Should we meet at your workplace or mine?" Now you just pick up the phone or run a video call directly, just call someone. It is incredibly more efficient. And it also means that you get entirely different contact surfaces, of course. (IP 10)

Some managers experienced that the new digital meeting form had made it easier for them to get to know their employees on a more personal level. It was described as if the threshold to get to know their employees had become lower and that digital meetings opened new topics of conversation when managers were, in a digital way, invited into their employees' home environments.

Hybrid work also presented some challenges in communicating with employees and colleagues. While some felt that reaching out to their colleagues was easier and more efficient, others experienced the opposite effect when the possibility of quickly discussing a question in the doorway with their colleagues was limited. The spontaneous interaction in the corridor was also perceived to have decreased after the transition to hybrid work when everyone was no longer present in the office at the same time. One effect of this was that it was more difficult for managers to stay updated on their employees' occupational well-being and health risks when they no longer spontaneously met them in the facilities. Some managers also reported that the challenge of being updated with occupational well-being and health risks was especially true concerning those employees who did not seek contact with the manager as much as others:

It is more difficult to see those with no one to have coffee with or be with. Those who may have felt left out before probably feel more left out now. And it is more challenging to pay attention and see everyone. And I think it's important that you are seen, and that there is someone who cares. (IP 6)

To avoid forgetting the occupational well-being and needs of certain employees in the new hybrid work arrangement, a new type of occupational safety and health management system was required from the managers.

That is a challenge now. How do I, as a busy manager, ensure that I do not just listen to those who are present when I am present? How do I listen to those who sit at home 50% of the time and maybe only come in one day when I am really busy or in meetings? (IP 13)

The managers described how check-ins often had a more informal character when everyone was working on site, while in hybrid work there is a greater need for managers to actively plan formal reviews and regular check-ins with their employees.

**New expectations for managers.** According to the interviewed managers, hybrid work has led to new expectations for the managerial role. One aspect of this was described as an increased expectation of being present in the office as a manager. On the one hand, some managers described how there was a perceived expectation from others in the organization that, as a manager, one should be available on-site when needed. On the other hand, some managers felt that there was a personal need for them as managers to be on-site to meet their employees. Compared to the time before the pandemic, some described it as a more limited freedom:

> I would probably like to have the opportunity to choose a bit more than I actually have. But it will not... It comes with the role, and other values I have. That my desire to work a bit more remotely, that is not going to happen. (IP 14)

Being present on-site as a manager was also related to the importance of setting an example. Managers described it as being a role model or, as described by one manager, as needing to "walk the talk" (IP 2) to show that there are advantages to working from the office that cannot be achieved if solely working from home, indicating that managers would prefer more on-site presence.

> There is an additional value in getting together, and I would like the employees to think so too and to consider that, yes, sure, I may be at home two days a week or a little more, but I choose to be there because I think there is value in it. (IP 8)

Inspiring through their actions rather than micromanaging their employees was also described as an essential component of leadership. Some managers found it was a tricky balancing act between granting employees' autonomy and the necessity to intervene and maintain control to ensure positive work outcomes and collaboration among colleagues.

Another aspect mentioned by managers was related to the employees' perceived boundaries between work and private life. While many managers acknowledged that hybrid work had improved their employees' work-life balance, they also recognized some challenges that emerged from it. Hybrid work has placed new demands on managers to pay attention to how employees allocate their working time. One manager described this as requiring a new type of responsiveness. The problem of boundary drawing was described as partly concerning the division between work and private life and partly concerning the division of work and breaks during the working day. Managers described an increased concern about the lack of recovery of some employees when working from home:

> We had to really push and say "you cannot work anymore. This is the end of the day. You are shutting down" and to be tough in that way. (IP 5).

Overall, the new expectations of the managerial role contributed to the managers needing to exchange ideas and experiences with other managers. Some managers described how they already had forums for such exchanges, while other managers wished for more organizational support in this regard:

> I need to think about it a bit first and actually figure out what I would want going forward. Well, maybe a bit more opportunity for interaction and dialogue with other managers. We've talked about this with HR to see if we could do something together. (IP14).

One of the key purposes of such forums for exchanging experience among managers could be, according to the managers, to learn together and find new solutions.

## Discussion

This study aimed to contribute to the literature on managers' experience of implementing hybrid work. The results show that while the managers reported several advantages of hybrid work, these benefits were often related to the employees within professional services. The managers generally discussed fewer benefits that could be directly linked to themselves and their managerial practices.

Our results show that managers regarded hybrid work as necessary to attract and retain skilled employees and as an enabler of new and more efficient working methods. The managers in our study also expressed a need to communicate and share experiences with other managers in the same situation, which aligns with the suggested implications from previous research [24]. The managers' perception was that hybrid working practices increased overall employee satisfaction, compared to the pre-pandemic situation and the mandatory work-from-home period during the pandemic. This perception aligns with previous research showing that satisfaction with teleworking is affected by its scope [21,40] and by personal preferences, which in turn may interact with life and career stages [8]. In line with other studies, our study suggests that hybrid work seems to enhance work-life balance [40] and help reduce the risks of professional isolation [17,21]. However, our study also suggests that the flexibility enabled by hybrid work does not extend to managers to the same degree as employees, since managers reported feeling a greater expectation to meet their employees regularly on-site. While hybrid work offers flexibility to employees, our study suggests that managers experience less of this benefit, reporting greater pressure to be on-site regularly and wishing their employees were on-site when they are. Managers' wishes for on-site work and leading by example in this case might also affect employees when they plan how much and when to work from home. This disparity, coupled with new managerial demands, risks employee neglect due to potentially infrequent manager-employee interactions.

Managers also experienced it as more challenging to help employees set boundaries between work and private life. Our results and previous studies [41] indicate that hybrid work might blur boundaries between work and leisure, which means that self-regulation becomes an essential skill during hybrid work [42]. Our results add, in this respect, to research on line managers and their importance for hybrid workers' occupational safety and health (OSH) [43].

Our main findings on the challenges experienced by the managers in the transition to hybrid work, including finding time to meet all employees at the office and follow their work related health, handling emotions among the staff regarding insecurities and expectations, and creating and strengthening the feeling of being a community at work, are in line with a previous Swedish study [7]. Previous research has identified several challenges for managers in hybrid work, such as their need to consider both organizational and employee needs, and that employees are a heterogeneous group whose preferences and needs concerning hybrid work might vary [23]. One recent study identified two groups of hybrid workers: one group that preferred to work from the office and one that preferred to work from home. The two groups differed in terms of the distribution of males/females, commuting time, and communicative work [44]. Based on our results, it seems vital to find new forms of communication to ensure that social exchanges are not lost, such as spontaneous meetings in corridors and coffee rooms. Investing in communication in teams and the organization is thus essential when introducing hybrid work [45]. When developing new ways of communicating, it is important to bear in mind that employees might experience too much managerial communication as intrusive, thus creating new demands on technical and social interaction skills for managers in hybrid workplaces [20,25]. In hybrid work, an important part of leadership, according to our results, is to build and maintain a sense of belonging to a group and an organization together with employees, also identified in previous research [45]. For this to succeed, a manager's ability to create trust is required [20], which is one aspect that has been identified as both particularly important [46] and challenging [47] in studies of leadership in digital contexts.

The fact that hybrid work can pose challenges for managers has also been seen in previous studies of telework during the pandemic [48]. The results of our study indicate that the challenges remain even in the transition to hybrid work. At the same time, the manager is often described as an important facilitator of the transition to hybrid work or more flexible

ways of working [22]. The current study shows the new expectations on the manager's role that such a transition may entail, such as leading by example and meeting the expectations of others, managing uncertainty and injustice, working with active measures to promote social interaction, encouraging social interactions at work, being sensitive to possible boundary problems between work and leisure time, formalizing check-ins with employees so that no one is at risk of being forgotten, and at the same time balancing between giving freedom and control.

## Strengths and limitations

One strength of the current study is the in-depth descriptions of managers' experiences of an ongoing transition to hybrid work, capturing their experiences of this process during the increase of hybrid work models implemented in knowledge-intensive workplaces following the COVID-19 pandemic. The current study also has limitations; while it is based on in-depth interviews capturing the experiences at one HEI, adding important new knowledge on hybrid work in this setting, its conclusions might not be generalisable to other contexts. Employees within HIEs often perform their work with high autonomy and discretion, and our results might therefore not be transferable to more controlled work contexts. Future research could benefit from including multiple HEIs and following managers' experiences over time. Future research could also benefit from a research design aiming to catch potential gender-related patterns relating to managers and their perception of hybrid work, by including specific questions in the interview guide and formally analysing differences and similarities between female and male managers or female and male-dominated organizations or sectors. Furthermore, there might be inherent limitations when studying one's own context and organisation. Even though there were no previous connections between the interviewers and interviewees, participants might be more prone to provide socially desirable responses when being interviewed by someone from the same organisation. To prevent bias due to social desirability, interviewers dedicated time before the start of each interview to inform participants about confidentiality and how the gathered data would be handled and reported [49].

## Conclusions

The shift to hybrid work requires managers to continuously manage employee expectations and worries, digitize work processes, and maintain or improve productivity. Managers have observed that hybrid work can improve work-life balance but also presents challenges in social interaction and employee relations. These opportunities and challenges necessitate adjustments in leadership, particularly in communication and managing workplace relationships, both within teams and across the organization. Despite these challenges, managers are generally optimistic about hybrid work and see it as the future. While these changes can be demanding, managers could benefit from increased support from the organization alongside networking and information exchange with other managers in similar situations.

## Author contributions

**Conceptualization:** Helena Tinnerholm Ljungberg, Martina Wallberg, Emmanuel Aboagye, Gunnar Bergström, Christina Björklund, Lydia Kwak, Susanna Toivanen, Irene Jensen.

**Data curation:** Helena Tinnerholm Ljungberg, Martina Wallberg, Emmanuel Aboagye, Gunnar Bergström, Christina Björklund, Lydia Kwak, Susanna Toivanen, Irene Jensen.

**Formal analysis:** Helena Tinnerholm Ljungberg, Martina Wallberg, Emmanuel Aboagye, Gunnar Bergström, Christina Björklund, Lydia Kwak, Susanna Toivanen, Irene Jensen.

**Investigation:** Helena Tinnerholm Ljungberg, Martina Wallberg, Emmanuel Aboagye, Gunnar Bergström, Christina Björklund, Lydia Kwak, Susanna Toivanen, Irene Jensen.

**Methodology:** Helena Tinnerholm Ljungberg, Martina Wallberg, Emmanuel Aboagye, Gunnar Bergström, Christina Björklund, Lydia Kwak, Susanna Toivanen, Irene Jensen.

 

**Project administration:** Helena Tinnerholm Ljungberg, Irene Jensen.

**Resources:** Helena Tinnerholm Ljungberg, Martina Wallberg, Emmanuel Aboagye, Gunnar Bergström, Christina Björklund, Lydia Kwak, Susanna Toivanen, Irene Jensen.

**Software:** Helena Tinnerholm Ljungberg, Martina Wallberg, Emmanuel Aboagye, Gunnar Bergström, Christina Björklund, Lydia Kwak, Susanna Toivanen, Irene Jensen.

**Supervision:** Helena Tinnerholm Ljungberg, Emmanuel Aboagye, Gunnar Bergström, Christina Björklund, Lydia Kwak, Susanna Toivanen, Irene Jensen.

**Validation:** Helena Tinnerholm Ljungberg, Irene Jensen.

**Visualization:** Helena Tinnerholm Ljungberg.

**Writing – original draft:** Helena Tinnerholm Ljungberg, Martina Wallberg, Irene Jensen.

**Writing – review & editing:** Martina Wallberg, Emmanuel Aboagye, Gunnar Bergström, Christina Björklund, Lydia Kwak, Susanna Toivanen, Irene Jensen.

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
