## [Decision Letter · Decision Letter 0]

17 Sep 2025

We look forward to receiving your revised manuscript.

Kind regards,

Mattia Vacchiano, Ph.D.

Academic Editor

PLOS ONE

Journal Requirements:

3. In the online submission form, you indicated that the datasets generated and analysed during the current study are not publicly available due to the Swedish ethical review regulation. Data are available upon reasonable request. Inquiries for data access should be sent to Karolinska Institutet, Institute of Environmental Medicine, Unit of Intervention and Implementation Research for Worker Health, Box 210, 171 77 Stockholm or contact the first author, helena.tinnerholm.ljungberg@ki.se, who will then contact the Swedish Ethical Review Authority for permission to openly share the data.

4. We note that there is identifying data in the Supporting Information file <Swedish Ethical Review Authority Decision 2021-03637.pdf and Decision 2021-03637 translated 2025-06-02.docx>. Due to the inclusion of these potentially identifying data, we have removed this file from your file inventory. Prior to sharing human research participant data, authors should consult with an ethics committee to ensure data are shared in accordance with participant consent and all applicable local laws.

-Location data

**Editor Comments:**

Your manuscript has received very positive reviews, and I too am convinced that it is good work. I would particularly make emphasis on the need to be more precise on the analysis of the narrative data. What data technique is used? Is it a qualitative content analysis? In my opinion, more details are needed.

If it helps, I also wrote a review in PLOS in which some issues regarding the role of managers and hybrid work emerged: https://journals.plos.org/plosone/article?id=10.1371/journal.pone.0305567

**Comments to the Author**

1. Is the manuscript technically sound, and do the data support the conclusions?

Reviewer #1: Yes

Reviewer #2: Yes

2. Has the statistical analysis been performed appropriately and rigorously?

Reviewer #1: N/A

Reviewer #2: N/A

3. Have the authors made all data underlying the findings in their manuscript fully available?

Reviewer #1: Yes

Reviewer #2: Yes

4. Is the manuscript presented in an intelligible fashion and written in standard English?

Reviewer #1: Yes

Reviewer #2: Yes

Reviewer #1: 1. Scientific Quality Assessment:

The manuscript presents rigorous scientific research with a justified qualitative approach. The study design respects established research criteria, and both sample selection and protocols are well-described, relevant, and follow academic standards. The different steps have been executed correctly. Apart from a few minor elements requiring clarification, the entire manuscript can be validated as is.

This manuscript makes a solid contribution to understanding hybrid work from the perspective of managers, which is particularly valuable given the relative scarcity of research focused on leadership experiences rather than employees experiences. The qualitative methodology is rigorous and appropriate, and the findings offer new post-pandemic insights. The research is particularly effective in highlighting the processual nature of adapting to hybrid work and the new challenges for managers in this transition.

2. Statistical Methods:

Although statistical methods are not applicable to this qualitative study, the analytical procedure complies with recognised standards for qualitative research. The authors demonstrate methodological rigour through systematic transcription, iterative coding with validation discussions, author triangulation and consensus-building processes.

3. Data Availability:

The data availability statement is appropriate and transparent for qualitative research. The authors clearly explain in the Additional Information that interview transcripts cannot be made publicly available due to national ethical review regulations, which aligns with ethical requirements for participant protection. The provision of a clear data access procedure (contact details for the Institute and first author, with explicit mention of the national Ethical Review Authority approval process) demonstrates commitment to research transparency within ethical constraints. While the detailed access procedure is provided to editors, a brief mention in the manuscript (e.g., "Data available upon reasonable request following ethical approval") could enhance reader transparency, though this is not essential.

4. Language

The manuscript demonstrates clear English expression and maintains a coherent organizational structure throughout. The presentation is generally accessible. Nothing to be addressed at this level.

5. Detailed Review Comments

Introduction

The research problem is well-articulated, and the theoretical positioning generally provides adequate foundation. Positive aspects include the innovative research focus on managerial versus employee perspectives, which is well-developed. Good utilization of pre- and post-pandemic literature is evident, and literature on managerial difficulties is well-referenced.

Several specific refinements would strengthen this section (listed in order of importance):

• Lines 52: Imprecision regarding employer initiative versus employee "own choice" according to the cited source (and literature in general). This may be an inadvertent error, but it requires verification.

• Lines 62-66: The assertion regarding multi-localized teams in the pre-pandemic context requires empirical support or should be presented with more nuance rather than as established fact. In my opinion, this claim is not self-evident, as literature had already demonstrated telework practices within companies localized in single locations.

• Lines 56-61: The paragraph defining "hybrid work" could be clearer with more sourcing to avoid potential confusion. While it is very clear in the rest of the manuscript, this section stating "Other terms such as full/partial telework, remote work, and work from home are also used interchangeably with hybrid work in some contexts" creates confusion. If partial telework can be used interchangeably, and sometimes remote work or work from home (which can be different types), this cannot be the case for "full telework" which by definition distinguishes itself from hybrid work.

• Lines 50-51: Trends in teleworking could benefit from more robust observational data sources. In my opinion, it would be advantageous to consider incorporating findings from articles such as Barrero, Bloom and Davies (2023) (https://doi.org/10.1257/jep.37.4.23) or similar longitudinal studies to ground these historical claims more firmly, rather than relying solely on overly specific thematic articles on telework.

General reflection: The particular status of Higher Education institutions might deserve somewhat more development regarding potential implications (versus sectors with micro-managing/increased surveillance) for the generalization of results.

Methods

The qualitative framework demonstrates appropriate alignment with the research objectives and complies with established standards for qualitative research, meeting recognised criteria as explained directly.

The interpretive design with semi-structured interviews is appropriate for the study objectives. The explanation of the study framework, including the context of the pandemic and the transition to hybrid situations, is adequately presented. Pre-tests, validation procedures, interview details, and ethical safeguards are documented. The manuscript mentions that informed consent and national ethical approval were obtained.

The diverse sampling strategy (gender, location, areas of interest) and transparency in reporting on the recruitment process (from invitations to the 15 final participants) demonstrate methodological rigour. The saturation of data obtained through iterative analysis is documented. Conventional content analysis with author triangulation provides adequate rigour for exploratory research.

Demographic variables are documented to demonstrate the diversity of the sample, which enhances the credibility of the qualitative results. The study focuses on shared managerial experiences rather than comparative analysis of subgroups. However, given the documented gender differences in telework experiences and managerial challenges in the existing literature, a brief reflection on possible gender-related patterns in the data, even if not formally analysed, could provide valuable context for future research directions. This is not a methodological requirement, but rather an opportunity to provide insights.

Results

The thematic organization is clear and well-supported by participant voices. The three main themes with six sub-categories provide a coherent analytical framework. Overall, there is good use of verbatim quotes and an appropriate balance between them and the argumentation.

Strengths:

• "Hybrid as ongoing process of change" (Line 258) is particularly well-developed, with excellent use of verbatim (Lines 267, 271) that illuminates the complexity of managerial adaptation

• The tension between transparency and employee privacy protection (Lines 272-279) is interesting and represents maybe an innovative contribution to literature.

• Recognition of managers' roles in maintaining social cohesion and organizational culture is well-articulated, as are the different new challenges they face

• The managers' role in actively encouraging work-life boundaries (Line 392), telling employees to take breaks or stop working, provides valuable managerial perspective on boundary management challenges previously documented from employee viewpoints

Areas for Development: Several nuanced observations merit deeper exploration:

• While the authors acknowledge managers' preference for on-site presence (Line 383-384), the potential tension between this preference and genuine hybrid flexibility could merit further discussion, particularly given the emphasis on managers 'leading by example' through physical presence (Line 380-87).

• The absence of discussion about micromanagement risks in telework settings represents a missed opportunity, given the literature's documentation of surveillance challenges and the counterproductive nature of excessive monitoring in remote contexts, although this is explained by the Higher Education institutional context.

• The discussion of enhanced communication accessibility could benefit from acknowledgment that increased managerial contact might be perceived as intrusive

• The final section on managers' need for peer exchanges and learning (Lines 403-404) introduces an important idea but lacks supporting verbatim and appears somewhat disconnected from the preceding analysis

Discussion

The discussion effectively contextualizes findings within existing literature and demonstrates awareness of study limitations.

Some claims would benefit from stronger empirical grounding, particularly the assertion that "hybrid work is most beneficial" (Line 417) which lacks systematic comparison with other arrangements. The manuscript addresses a hybrid telework situation and its challenges but does not really provide elements that could allow comparison with a fully remote situation.

The alignment with literature is well-done in this section. The post-pandemic contextualization adds valuable temporal specificity to previous research.

The manuscript appropriately identifies study strengths and limitations, including adequate consideration of potential social desirability bias. However, as noted in previous sections, the Higher Education institutional context and its implications for generalizability could be addressed more thoroughly. (Lines 467-470) While the authors acknowledge the HE institutions context, deeper reflection on generalizability limitations would strengthen the contribution. Higher education institutions typically operate with high employee autonomy and minimal surveillance compared to many sectors, potentially limiting transferability of findings to more controlled work environments where increased supervision and monitoring practices may significantly alter the hybrid work experience.

Conclusion

Lines 484-485: The reported managerial optimism about hybrid work's future, which appears to be a new idea not reflected in the results, seems to contradict other conclusions regarding preferences for physical presence and certain persistent technical difficulties (hybrid meetings). This tension deserves to be explicitly acknowledged and interpreted from a theoretical perspective. This optimism on the part of managers does not appear to be evident from reading this manuscript.

The remainder of the conclusion appropriately synthesizes the study's contributions and implications.

**Reviewer #2: ** Thank you for the opportunity to review your manuscript, "Wish You Were Here": Managers’ Experiences of Hybrid Work in Higher Education. As you rightly point out, this is a timely and under-researched area, and your work contributes to a growing body of knowledge on the evolving nature of management in hybrid work environments.

Below are some suggestions that may help strengthen the manuscript:

Introduction

The theoretical background is solid; however, the manuscript would benefit from a deeper problematization of the specific setting, higher education. How does managing in a university or higher education context differ from management in other public sector workplaces? Why is it particularly important to examine hybrid work from this perspective? Clarifying this could better establish the significance of your research focus.

Design

The interpretative approach is appropriate, but it is currently only briefly referenced. The manuscript could be improved by elaborating on the methodological choices. For instance, why was this method particularly suitable for your setting and research questions? Including more concrete examples of how the method was applied would also enhance transparency and strengthen the rigor of your analysis.

Participants

As a reader, I would appreciate more contextual information about the participating managers. How many subordinates do they supervise? Are they first-line managers, middle managers, or in more senior leadership roles? This information would help readers better understand the perspectives represented in your findings.

Results

The main categories and subcategories are clearly presented and easy to follow. The structure works well in guiding the reader through your analysis.

Discussion

While this is optional, I believe the discussion could benefit from a more thorough engagement with all three main categories. You have explored the leadership implications in some depth, which is helpful, but further reflection on the other main findings, such as the broader implications of working in new ways and potential impacts on employees could enhance the discussion and provide a more holistic interpretation of the results.

**Do you want your identity to be public for this peer review?** For information about this choice, including consent withdrawal, please see our Privacy Policy

Reviewer #1: **No****: **

Reviewer #2: **Yes: ** Pär Löfstrand, Department of Psychology and Social Work, Mid Sweden University

---

## [Author Response · Author response to Decision Letter 1]

28 Oct 2025

Dear Editors,

We thank the reviewers and the editor for their positive and constructive comments on our initial submission. In response, we have revised the manuscript to improve readability and clarity and developed the text based on the reviewers’ and editor’s comments. We have revised parts of the introduction and further developed our discussion and added a more developed section on the context of Higher Education Institutions (HEIs). Our point-by-point responses are detailed below. References to lines in our response refer to the version of the manuscript without tracked changes.

***

Editor Comments:

Your manuscript has received very positive reviews, and I too am convinced that it is good work. I would particularly make emphasis on the need to be more precise on the analysis of the narrative data. What data technique is used? Is it a qualitative content analysis? In my opinion, more details are needed.

- Thank you for this opportunity to revise our manuscript, we have further developed the sections describing the Conventional content analysis used in the analysis on lines 205-213.

If it helps, I also wrote a review in PLOS in which some issues regarding the role of managers and hybrid work emerged: https://journals.plos.org/plosone/article?id=10.1371/journal.pone.0305567

- Thank you for this helpful comment. We have read and referred to the review in the introduction and discussion. (see lines 56, 60, and 467)

***

Comments to the Author

1. Is the manuscript technically sound, and do the data support the conclusions?

Reviewer #1: Yes

Reviewer #2: Yes

2. Has the statistical analysis been performed appropriately and rigorously?

Reviewer #1: N/A

Reviewer #2: N/A

3. Have the authors made all data underlying the findings in their manuscript fully available?

Reviewer #1: Yes

Reviewer #2: Yes

4. Is the manuscript presented in an intelligible fashion and written in standard English?

Reviewer #1: Yes

Reviewer #2: Yes

5. Review Comments to the Author

Reviewer #1: 1. Scientific Quality Assessment:

The manuscript presents rigorous scientific research with a justified qualitative approach. The study design respects established research criteria, and both sample selection and protocols are well-described, relevant, and follow academic standards. The different steps have been executed correctly. Apart from a few minor elements requiring clarification, the entire manuscript can be validated as is.

This manuscript makes a solid contribution to understanding hybrid work from the perspective of managers, which is particularly valuable given the relative scarcity of research focused on leadership experiences rather than employees experiences. The qualitative methodology is rigorous and appropriate, and the findings offer new post-pandemic insights. The research is particularly effective in highlighting the processual nature of adapting to hybrid work and the new challenges for managers in this transition.

2. Statistical Methods:

Although statistical methods are not applicable to this qualitative study, the analytical procedure complies with recognised standards for qualitative research. The authors demonstrate methodological rigour through systematic transcription, iterative coding with validation discussions, author triangulation and consensus-building processes.

3. Data Availability:

The data availability statement is appropriate and transparent for qualitative research. The authors clearly explain in the Additional Information that interview transcripts cannot be made publicly available due to national ethical review regulations, which aligns with ethical requirements for participant protection. The provision of a clear data access procedure (contact details for the Institute and first author, with explicit mention of the national Ethical Review Authority approval process) demonstrates commitment to research transparency within ethical constraints. While the detailed access procedure is provided to editors, a brief mention in the manuscript (e.g., "Data available upon reasonable request following ethical approval") could enhance reader transparency, though this is not essential.

-Thank you for drawing our attention to this section, which will (in line with all articles published in PlosOne) be printed on the first page of the article (in the left-most column), and we have therefore not included comments on this in the main text. However, we have made a slight change to the text to be in accordance with the Swedish ethical review regulation:

The datasets generated and analysed during the current study are not publicly available due to the Swedish ethical review regulation. Inquiries for data access should be sent to Karolinska Institutet, Institute of Environmental Medicine, Unit of Intervention and Implementation Research for Worker Health, Box 210, 171 77 Stockholm or contact the first author, helena.tinnerholm.ljungberg@ki.se, who will then contact the Swedish Ethical Review Authority.

4. Language

The manuscript demonstrates clear English expression and maintains a coherent organizational structure throughout. The presentation is generally accessible. Nothing to be addressed at this level.

5. Detailed Review Comments

Introduction

The research problem is well-articulated, and the theoretical positioning generally provides adequate foundation. Positive aspects include the innovative research focus on managerial versus employee perspectives, which is well-developed. Good utilization of pre- and post-pandemic literature is evident, and literature on managerial difficulties is well-referenced.

Several specific refinements would strengthen this section (listed in order of importance):

• Lines 52: Imprecision regarding employer initiative versus employee "own choice" according to the cited source (and literature in general). This may be an inadvertent error, but it requires verification.

- Thank you for noticing this. This section (lines 47-56) has been rewritten to increase clarity. Our point here is to illustrate the difference between the national policy of mandatory work from home during the COVID-19 pandemic versus teleworking before the pandemic, which could, at least in part, be responsive to worker or organizational preferences. New references have also been added to this section.

• Lines 62-66: The assertion regarding multi-localized teams in the pre-pandemic context requires empirical support or should be presented with more nuance rather than as established fact. In my opinion, this claim is not self-evident, as literature had already demonstrated telework practices within companies localized in single locations.

- Thank you, this section (lines 64-71) has been nuanced and developed, and new references have been added.

• Lines 56-61: The paragraph defining "hybrid work" could be clearer with more sourcing to avoid potential confusion. While it is very clear in the rest of the manuscript, this section stating "Other terms such as full/partial telework, remote work, and work from home are also used interchangeably with hybrid work in some contexts" creates confusion. If partial telework can be used interchangeably, and sometimes remote work or work from home (which can be different types), this cannot be the case for "full telework" which by definition distinguishes itself from hybrid work.

- Thank you for drawing our attention to this. Lines 58-63 have been rephrased, and a new reference has been added.

• Lines 50-51: Trends in teleworking could benefit from more robust observational data sources. In my opinion, it would be advantageous to consider incorporating findings from articles such as Barrero, Bloom and Davies (2023) (https://doi.org/10.1257/jep.37.4.23) or similar longitudinal studies to ground these historical claims more firmly, rather than relying solely on overly specific thematic articles on telework.

- Thank you for the comment and the helpful reference. Lines 47-57 have been rewritten, and Barrera et al is now referred to in this section.

General reflection: The particular status of Higher Education institutions might deserve somewhat more development regarding potential implications (versus sectors with micro-managing/increased surveillance) for the generalization of results.

- To answer this comment, as well as a comment from reviewer 2, the section starting on line 119 and running to line 139, has been developed to further describe the specific context for Higher Education institutions (HEIs).

Methods

The qualitative framework demonstrates appropriate alignment with the research objectives and complies with established standards for qualitative research, meeting recognised criteria as explained directly.

The interpretive design with semi-structured interviews is appropriate for the study objectives. The explanation of the study framework, including the context of the pandemic and the transition to hybrid situations, is adequately presented. Pre-tests, validation procedures, interview details, and ethical safeguards are documented. The manuscript mentions that informed consent and national ethical approval were obtained.

The diverse sampling strategy (gender, location, areas of interest) and transparency in reporting on the recruitment process (from invitations to the 15 final participants) demonstrate methodological rigour. The saturation of data obtained through iterative analysis is documented. Conventional content analysis with author triangulation provides adequate rigour for exploratory research.

Demographic variables are documented to demonstrate the diversity of the sample, which enhances the credibility of the qualitative results. The study focuses on shared managerial experiences rather than comparative analysis of subgroups. However, given the documented gender differences in telework experiences and managerial challenges in the existing literature, a brief reflection on possible gender-related patterns in the data, even if not formally analysed, could provide valuable context for future research directions. This is not a methodological requirement, but rather an opportunity to provide insights.

-We have added a short paragraph in the “strengths and limitations” section, opening for future research regarding gender-related patterns, managers, and hybrid work (lines 525-529).

Results

The thematic organization is clear and well-supported by participant voices. The three main themes with six sub-categories provide a coherent analytical framework. Overall, there is good use of verbatim quotes and an appropriate balance between them and the argumentation.

Strengths:

• "Hybrid as ongoing process of change" (Line 258) is particularly well-developed, with excellent use of verbatim (Lines 267, 271) that illuminates the complexity of managerial adaptation

• The tension between transparency and employee privacy protection (Lines 272-279) is interesting and represents maybe an innovative contribution to literature.

• Recognition of managers' roles in maintaining social cohesion and organizational culture is well-articulated, as are the different new challenges they face

• The managers' role in actively encouraging work-life boundaries (Line 392), telling employees to take breaks or stop working, provides valuable managerial perspective on boundary management challenges previously documented from employee viewpoints

Areas for Development: Several nuanced observations merit deeper exploration:

• While the authors acknowledge managers' preference for on-site presence (Line 383-384), the potential tension between this preference and genuine hybrid flexibility could merit further discussion, particularly given the emphasis on managers 'leading by example' through physical presence (Line 380-87).

- Thank you for providing this opportunity. A new sentence has been added to lines 473-477.

• The absence of discussion about micromanagement risks in telework settings represents a missed opportunity, given the literature's documentation of surveillance challenges and the counterproductive nature of excessive monitoring in remote contexts, although this is explained by the Higher Education institutional context.

- We agree with the comment and with the statement that the HIE context most probably explains the lack of micromanagement discussion amongst our interviewed managers. To make sure that this context is explained to our readers, we have further developed on the HIE context in the introduction, see lines 119-139.

• The discussion of enhanced communication accessibility could benefit from acknowledgment that increased managerial contact might be perceived as intrusive

- A new sentence on the risks of too much managerial communication has been added on lines 496-499.

• The final section on managers' need for peer exchanges and learning (Lines 403-404) introduces an important idea but lacks supporting verbatim and appears somewhat disconnected from the preceding analysis

- This section has been rephrased and a verbatim added on lines 447-450.

Discussion

The discussion effectively contextualizes findings within existing literature and demonstrates awareness of study limitations.

Some claims would benefit from stronger empirical grounding, particularly the assertion that "hybrid work is most beneficial" (Line 417) which lacks systematic comparison with other arrangements. The manuscript addresses a hybrid telework situation and its challenges but does not really provide elements that could allow comparison with a fully remote situation.

- Thank you for this helpful comment. To strengthen this claim, we have added a sentence in the “setting and procedure” section, explaining that the interviewed managers shared the experience of having transitioned from almost exclusively working on site to working from home during the COVID-19 pandemic to the new hybrid working arrangements after the pandemic (lines 151-153). We have also rephrased a section in the discussion, and we have also discussed these results with the help of new references (see lines 462-468). In this section, it has been clarified first that the interviewee managers see advantages with hybrid work when they compare this with previous ways of working, but in a following sentence discussing the benefits of hybrid work, the comparison with full telework has been removed.

The alignment with literature is well-done in this section. The post-pandemic contextualization adds valuable temporal specificity to previous research.

The manuscript appropriately identifies study strengths and limitations, including adequate consideration of potential social desirability bi

---

## [Editor Report · Decision Letter 1]

4 Nov 2025

Dear Dr. Tinnerholm Ljungberg,

First, I suggest some clarifications in the abstract. It is true that telework research has generally placed more emphasis on employees—yet managers are also workers, and the current phrasing of that part of the abstract could be somewhat misleading. I would also recommend concisely describing the sample in the abstract: how many participants were interviewed, who these managers are, and the sector or tasks involved. Please be explicit that this is a sample of managers employed in different services at a Swedish medical university.

At present, the abstract remains unchanged from the original version and could be more informative. Rather than focusing mainly on the analytical categories emerging from the narrative data—which are quite common in similar analyses—it would be ideal to include a concise summary of the main findings. As this is a qualitative study, we expect more nuance, new mechanisms, and a finer-grained understanding of these managers’ experiences.

Regarding the analytical technique, I am not familiar with the term “conventional content analyis.” It seems to me that it would be more appropriate to frame this as a Qualitative Content Analysis (QCA) (see Hsieh HF, Shannon SE. Three approaches to qualitative content analysis. Qual Health Res. 2005;15(9):1277–1288. doi:10.1177/1049732305276687; PMID: 16204405). In any case, it would be useful to add a methodological reference clarifying the data analysis.

Kind regards,

Mattia Vacchiano, Ph.D.

Academic Editor

PLOS ONE
---

## [Author Response · Author response to Decision Letter 2]

26 Nov 2025

PONE-D-25-27056R1

‘Wish you were here’: Managers’ experiences of hybrid work in higher education

PLOS ONE

Dear Editor,

Thank you for your positive response and constructive comments on our resubmission. We have now accordingly revised the manuscript. Our detailed point-by-point responses, addressing the remaining concerns, are provided below.

First, I suggest some clarifications in the abstract. It is true that telework research has generally placed more emphasis on employees—yet managers are also workers, and the current phrasing of that part of the abstract could be somewhat misleading. I would also recommend concisely describing the sample in the abstract: how many participants were interviewed, who these managers are, and the sector or tasks involved. Please be explicit that this is a sample of managers employed in different services at a Swedish medical university.

- Thank you for providing us with this opportunity. We have rephrased the mentioned sentence in the abstract to avoid (unintentionally) suggesting that managers are not employees (please see lines 31-32). Further clarifications on the participants and the setting have also been added to the abstract on lines 32-35.

At present, the abstract remains unchanged from the original version and could be more informative. Rather than focusing mainly on the analytical categories emerging from the narrative data—which are quite common in similar analyses—it would be ideal to include a concise summary of the main findings. As this is a qualitative study, we expect more nuance, new mechanisms, and a finer-grained understanding of these managers’ experiences.

- The summary of the conclusions is further developed in the abstract (lines 40-48) to highlight that this study provides a more fine-grained understanding of managers' experiences.

Regarding the analytical technique, I am not familiar with the term “conventional content analyis.” It seems to me that it would be more appropriate to frame this as a Qualitative Content Analysis (QCA) (see Hsieh HF, Shannon SE. Three approaches to qualitative content analysis. Qual Health Res. 2005;15(9):1277–1288. doi:10.1177/1049732305276687; PMID: 16204405). In any case, it would be useful to add a methodological reference clarifying the data analysis.

- We have rephrased the abstract (line 38) and the methods section (lines 215-224) to clarify that the method used is a qualitative content analysis (QCA). However, following Hsieh and Shannon (2005, p. 1277), we also think it is important to show which of the three distinct main approaches they list (conventional, directed, and summative) we have used, namely “the conventional content analysis approach”. We have clarified this by consistently using the word 'approach' when introducing and referring to “the conventional content analysis approach,” and by adding the reference again when describing how it has structured the data analysis (lines 215-224).

Finally, the title is attractive—and although I am a Pink Floyd fan—the message it conveys seems to suggest that managers are unhappy with hybrid work. Your results shows many challenges but appear to be more nuanced. I am ok with keeping the title, but I would therefore recommend justifying it more explicitly in the discussion or adjusting it to better reflect your findings.

- Thank you for giving us this opportunity to develop our results, further nuance managers' experiences, and keep the title. To further illustrate managers’ wish for on-site presence to facilitate their own leadership roles and tasks, while simultaneously appreciating the benefits their employees enjoy in hybrid work, subsequent changes have been made in the discussion (481-484) and in the abstract (lines 46-48).

---

## [Editor Report · Decision Letter 2]

2 Dec 2025

‘Wish you were here’: Managers’ experiences of hybrid work in higher education

PONE-D-25-27056R2

Dear Helena Tinnerholm Ljungberg,

We’re pleased to inform you that your manuscript has been judged scientifically suitable for publication and will be formally accepted for publication once it meets all outstanding technical requirements.

Kind regards,

Mattia Vacchiano, Ph.D.

Academic Editor

PLOS ONE

---

## [Editor Report · Acceptance letter]

PONE-D-25-27056R2

PLOS One

Dear Dr. Tinnerholm Ljungberg,

I'm pleased to inform you that your manuscript has been deemed suitable for publication in PLOS One. Congratulations! Your manuscript is now being handed over to our production team.

Kind regards,

on behalf of

Dr. Mattia Vacchiano

Academic Editor

PLOS One